# New Challenges for Biobanks: Accreditation to the New ISO 20387:2018 Standard Specific for Biobanks

**DOI:** 10.3390/biotech10030013

**Published:** 2021-07-02

**Authors:** Pasquale De Blasio, Ida Biunno

**Affiliations:** Integrated Systems Engineering Srl, c/o OpenZone, Via Meucci 3, 20091 Bresso, MI, Italy; ida.biunno@irgb.cnr.it

**Keywords:** biobanks, certification, accreditation, quality management systems, QMS, ISO 9001:2015, ISO 20387:2018, biobanking engineering

## Abstract

Background: The emergence of “multi-omics” and “multi-parametric” types of analysis based on a high number of biospecimens enforces the use of a great number of high-quality *“Biological Materials and Associated Data”* (BMaD). To meet the demands of biomedical research, several Biological Resource Centers (BRCs) or Biobanks world-wide have implemented a specific Quality Management System (QMS) certified ISO 9001:2015 or accredited by CAP^9^ ISO 20387:2018. For the first time, ISO, with the support of several Biobanking experts, issued the ISO 20387:2018 which is the first ISO norm specific for Biobanks. The fundamental difference with present certification/accreditation standards is that the ISO 20387:2018 focuses not only on the operational aspects of the Biobank, but also on the “competence of the Biobank to carry our specific Biobanking tasks”. Methods: The accreditation process for ISO 20387:2018 required the definition of: (1) objectives, goals and organizational structure of the Biobank, including procedures for governance, confidentiality and impartiality policies; (2) standard operating procedures (SOPs) of all activities performed, including acquisition, analysis, collection, data management, distribution, preparation, preservation, testing facility and equipment maintenance, calibration, and monitoring; (3) procedures for control of documents and records, the identification of risks and opportunities, improvements, corrective actions, nonconforming records and evaluation of external providers (4) an internal audit and management reviews, verification of QMS performance, monitoring of quality objectives and personnel qualification and competency in carrying out specific Biobanking tasks. Results: The accreditation process is performed by an independent authorized organization which certifies that all processes are performed according to the QMS, and that the infrastructure is engineered and managed according to the GDP and/or GMP guidelines. Conclusion: Accreditation is given by an accreditation body, which recognizes formally that the Biobank is “competent to carry out specific Biobanking tasks”.

## 1. Introduction

According to the OECD [1,2] (Organization for Economic Co-operation and Development), “Biological Resource Centers (BRCs) are an essential part of the infrastructure underpinning life sciences and biotechnology. These consist of service providers and repositories (“Biobanks”) of living cells, genomes and information relating to the heredity and functions of biological systems. BRCs contain collections of culturable organisms (e.g., micro-organisms, plant, animal and human cells), replicable parts of these (e.g., genomes, plasmids, viruses, DNAs), viable but not yet culturable organisms, cells and tissues, as well as databases containing molecular, physiological and structural information relevant to the collections and related bioinformatics.” Biobanks must meet the high standards of quality and expertise demanded by the international community of scientists and industry for the delivery of biological information and materials. They must provide access to biological resources on which R&D in the life sciences and the advancement of biotechnology depends. In this paper, the word “Biobank” is used, for simplicity, to describe the accreditation process for a “BRC and or Repository”.

The precise definition of what constitutes a Biobank is still controversial and the scientific community has not as of yet accepted any universal definition. However, it is important to clearly define the term since it is an important step towards fostering collaboration amongst researchers, facilitating the identification of samples’ potential sources. 

ISO 20387:2018 [3] defines a Biobank as “a legal entity or part of a legal entity that performs Biobanking”, and the term Biobanking as “the process of acquisitioning and storing, together with some or all of the activities related to collection, preparation, preservation, testing, analyzing and distributing defined biological material as well as related information and data” (BMaD).

The challenges for Biobanks are to move forward from the certification scheme (ISO 9001:2015 [4] to the ISO 20387:2018 [3] accreditation scheme. The new challenges are to change the focus of the Biobank from strictly operational aspects (e.g., QMS, SOPs, technical instructions) to governance and management aspects such as personnel performing Biobanking activities on behalf of the Biobank, as well as the entity itself. Generally, a Biobank is an organization in which “Biospecimens and Associated Data” (BMaD) are collected, processed, stored and distributed for use in clinical care and for research purposes [5,6,7]. A Biobank collects a wide range of specimens which include a collection of different types of materials such as human, non-human (e.g., microbes), animals and plants^1^. Initially, Biobanks were found in universities which over time grew into government and institutional storing centers with biospecimens used commercially and for research purposes.

In the beginning, these infrastructures were collecting only basic data such as the diagnosis and date of collection; today, a wide range of information is required, including several aspects of patient and participant phenotypes in addition to proteomic, genetics and other information under the “omics” category with the objective to serve as a fundamental tool for personalized medicine [1,8]. Today, Biobanks can perform acquisition, processing and storage of BMaD for not-yet-identified future use(s). In these cases, the Biobank can collect the BMaD according to “*informed consent*” following standard operating procedures (SOPs) appropriate for the projected use of the BMaD. Alternatively, Biobanks can collect BMaD in response to a request from a user for a specific purpose/project. Biobanks can acquire BMaD for investigators studying new methods of collecting, storing, or processing biological materials and the effects of these new methods on various analytes. In these cases, the Biobank can tailor the procedures to specifically meet the investigator’s needs rather than following widely accepted SOPs for handling of the BMaD.

Biobanks cannot vary only on the type of samples to collect and store but also on the type of activities that can be performed according to the international guidelines [1,2,9,10] and by the definition of Biobanking in ISO 20387:2018 [3], (i.e., collection/acquisition, preparation, preservation, testing, storage, analyzing and distribution of BMaD) or a subset of these activities (i.e., collecting and distributing). Biobanks can involve different types of organizations; they can be independent legal entities or reside within governmental entities, academic institutions, hospitals, non-profit or commercial organizations. Biobanks can include multiple operation sites and can sometimes involve parties working in multiple institutions or organizations. In addition, they can involve sites of operations within different regions or sometimes even different countries. In this case, the quality of the biospecimen can vary according to the SOPs used or during shipment of the biospecimen from the collection site to the central biobank. In this case, it is important that all collection sites use the same SOPs for collection, packing and shipment. It is also very important to associate clinical data (with a standardized minimum dataset) to the biospecimen up-loaded to the central LIMS for the project. Pre-analytical data should also be collected according to SPIDIA guidelines [10], even if there is a lack of knowledge among the Biobanking sector on the importance to collect pre-analytical data. 

It is up to the Biobank to identify the scope of Biobank activities for which it wants to be accredited ISO 20387:2018 [3]. In the last decade, several Biobanking organizations (ISBER, ESBB, OECD, IARC, CTRnet, BBMRI-Eric) [1,2,8,10,11,12] have published guidelines for the Biobank operations and become a part of international networks (e.g., NCI, CTRnet, BBMRI-ERIC, etc.) [1,2,8,10,11,12] which are actively engaged in harmonization activities between the Biobank of the network, promoting and contributing also to the definition of the ISO 20387:2018 [3] standards developed specifically for Biobanks.

## 2. Discussion and Results

Biological resources are an essential material for basic and translational research as well as for clinical studies [5,6,7,8,9,10,11,12,13]. The “omics” technologies have made unprecedented progress thanks to the availability of “high quality biospecimens” and have contributed to a better understanding of human health and diseases. The development of personalized medicine [13,14] was made possible by the developments of genomic and proteomic platforms, molecular imaging and bioinformatics. Most of our current knowledge on diseases is based on the systematic investigation of human biological samples and medical data. Quality of samples [15,16,17] can be the bottleneck for diagnosis and research studies in case the BMaD [10] are collected by several Biobanks which use different standards [10,15,18]. The collection of samples, transport, process storage and distribution are a very complex process and must be studied and standardized by the Biobanks in harmony with the research centers taking part in the study. The use of automation and a robotic system, must also be evaluated in order to standardize sample preparation, increase throughout and improve the quality of the process [15,18]. ISO TR22758 [8] (implementation guideline for ISO 20387:2018 [3]), expands selected aspects of ISO 20387 [3] in particular the concept of “fit for purpose” to determine the suitability of the Biobanking outputs, scope of application/conformity and competency of Biobanking personnel. ISO 20387:2018 [3] and ISO TR 22758 [8] are the first ISO standards focused on Biobanking management. Implementation of the ISO 20387:2018 [3] will impact positively on the future of the Biobank, improving the level of trust between the Biobanks, donors, researchers and regulatory bodies. By undergoing third-party assessments, Biobanks can demonstrate compliance to these standards and increase confidence in their users. The scope of the ISO 20387:2018 [3] is to: (1) improve access to qualified samples and data (BMaD), (2) ensure/increase the quality of samples (implementing Quality Management System (QMS) and Quality Control), (3) unify policies and procedures for the Biobank, (4) support exchange of biological material and related data among Biobanks and researchers, (5) stimulate access of partners from public and private sector, (6) increase stakeholder confidence and assurance, (7) be applicable for Biobanks of different sizes, (8) foster R&D, (9) reduce costs (due to a lack of reproducibility of scientific studies) [5,15,16,17,18] and research waste (by improving Biobanking operation). To achieve the ISO 20387:2018 [3] accreditation, the Biobank needs to develop a comprehensive QMS specifically designed for the establishment and operation of the Biobank. The QMS should be established around the following main activities:

1. **Biobanking governance** must define the objectives and the organizational structure of the Biobank, including procedures for governance, confidentiality and impartiality policies, with a clear line of authorities, responsibility for each reporting line and accountability for each role. In addition, a set of organizational procedures, such as the control of documents and records, identification of risks and opportunities, verification of QMS performance, continuous improvements, corrective actions, and nonconforming records, must be established and periodically reviewed during the “Internal Audit” and “Management Reviews” which must also review “quality objectives” and “personnel qualification and competency” which are also important for the accreditation process. Figure 1 below describes the key elements to consider in the ISO 20387:20183 accreditation process: 

2. **Biobank processes**. As indicated before, there are different types of Biobanks which can process different types of biospecimens with different methods and logistics. Figure 2 below, shows an example of a possible Biobanking workflow with the different types of topics, processes and activities. It is up to the Biobank to identify the activities and the best workflow. Each Biobank must define a QMS and Standard Operating Procedure (SOP) or technical documents for each process, which must describe in detail the steps to follow by authorized and qualified personnel. Each SOP (if required) should indicate quality indicators and acceptance criteria, which must be recorded each time the SOPs are executed. The “Quality Indicators” are important to set quality standards and objectives for the Biobank in order to be able to plan continuous improvement programs. The personnel should be trained and qualified and authorized to execute the SOPs.

3. **Biobank infrastructure** (Figure 3 below) must be engineered with the objective to follow the most stringent International guidelines and comply with local laws and regulations, require the use of safety procedures (including the use of Personal Protective Devices (PPE) for all biological processes and cryo-storage activities, obtain validation (from a third party) of all critical equipment (such as the LN2 distribution system, vessels and containers, mechanical −150 °C/−80 °C ULT freezers and −40 °C/−20 °C freezers, monitoring and alarm system, Biobank LIMS), to establish with specialized vendors preventive maintenance contracts for all Biobank critical equipment and instruments, appoint on-duty personnel for alarm intervention 24/7, and establish a Disaster Recovery Plan.

After completing the QMS documentation, it is important to prove that the Biobank personnel is well trained and competent in using the SOPs of all the processes described in the workflow (Figure 1 and Figure 2). This process requires 3–6 months of operations with the execution of internal audits to verify the competence of the personnel and use of SOPs for all activities of the Biobank. At the end of these activities, the Biobank is ready to have the accreditation visit (external audit) by the accredited body.

## 3. Material and Methods

Considering that many Biobanks are ISO 9001:2015 [4] certified or CAP accredited [6], the process to get accredited to the ISO 20387:2018 [3], should start with a gap analysis between the new ISO norm or CAP, with the objective to review the QMS, including the specific requirements of the ISO 20387:2018 [3], focusing on: (1) Biobanking Governance, in particular the Biobank organization, personnel training, qualification and competence, etc.; (2) Biobanking processes reviewing SOPs, quality indicators, quality improvements, preventive and corrective actions, etc.; (3) Biobanking infrastructure, with particular attention to the engineering aspects of the Biobank, looking at safety procedures, monitoring and alarm system, construction aspects (e.g., air conditioning and air circulation system, back-up system, etc.).

At the end of the gap analysis, it is important to define a project plan (GANT chart) with the identification of all activities necessary to improve the QMS, including the ISO 20387 [3] requirements. The project plan should be managed by the Biobank quality manager (or a specific project manager) which, with the collaboration of the Biobank management and staff, should implement all required activities. When the QMS is reviewed and approved by the Biobanking management, the personnel should be trained and qualified to perform specific processes according to their competences. Each process (SOP) should be executed and recorded, and quality objective should be set. This process usually takes a few months which is necessary to validate all critical equipment (e.g., LN2 containers, −80 °C ULT Freezers, etc.), test the monitoring and alarm systems, implementing the safety procedure for the low-oxygen alarm, electric power failure, disaster plan, etc. During this phase, internal audits must be performed in order to detect deviations in respect to the ISO 20387 [3] requirements, define preventive and corrective actions, set a quality objective, etc. The Biobank at this point is ready for the ISO 20387:2018 [3] audit performed by the independent authorized organization.

## 4. Conclusions

The objectives for a Biobank to get accredited to ISO 20387:2018 [3] are: (1) improve the access to qualified samples and data (BMaD); (2) ensure/increase the quality of samples (implementing QMS and quality control); (3) unify policies and procedures for the Biobank; (4) support exchange of biological material and related data among Biobanks and researchers; (5) stimulate access of partners from the public and private sector; (6) increase stakeholder confidence and assurance; (7) be applicable for Biobanks of different sizes; (8) foster R&D; (9) reduce costs (due to the lack of reproducibility of scientific studies) [5,15,16,17,18] and research waste (by improving Biobanking operation). To achieve the ISO 20387:2018 [3] accreditation, the Biobank needs to review the QMS, including the ISO 20387:2018 [3] requirements. The process requires dedicated skilled resources, time (between 6 and 12 months), and substantial investment (EUR 15–25 K for the accreditation process).

Accreditation to the ISO 20387:2018 [3] will impact positively on the future of the Biobank improving the level of trust between the Biobanks, donors, researchers and regulatory bodies. By undergoing third-party assessments, Biobanks can demonstrate compliance to these standards and increase confidence in their users. Due to the increase in complexity and requirements of basic and clinical research, accredited Biobanks to ISO 20387:2018 [3] will have a competitive advantage among non-accredited Biobanks.

The effort and cost must be considered to be accredited with ISO 20387:2018 [3]. The Biobank management must carefully consider the “costs and benefits” and create a specific project for the “Accreditation Process”, which should have a specific budget and dedicated resources. Depending on the size of the Biobank, the timing to complete the preparation phase (e.g., review and update to the new standards the (1) governance, (2) Biobank and workflow, (3) infrastructure, (4) personnel training and competence review) takes usually from 6 to 12 months. The costs are mainly personnel effort (2–4 man/months), additional training courses to improve personnel competency and, if needed, infrastructure improvements. Accreditation costs must also be considered, which could vary from EUR 15 K to EUR 25 K (depending on the “accreditation Body”) and additional 8–10 K EUR/year for maintaining the accreditation which lasts 4 years.

## Figures and Tables

**Figure 1 biotech-10-00013-f001:**
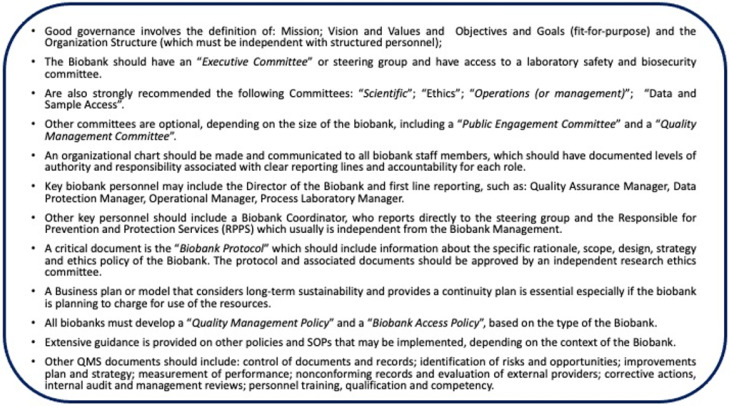
Biobank governance the key elements.

**Figure 2 biotech-10-00013-f002:**
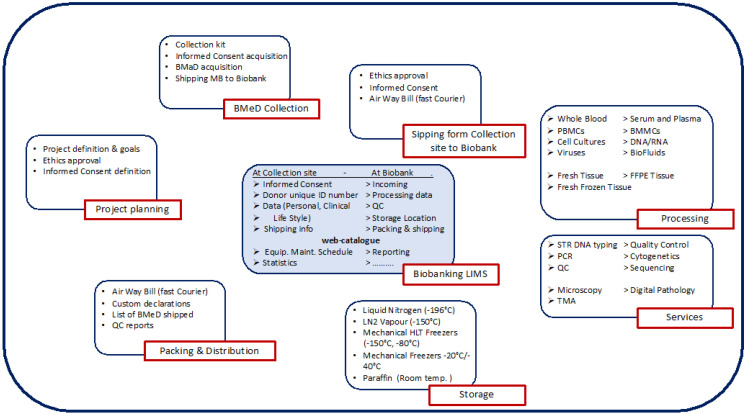
Biobanking workflow.

**Figure 3 biotech-10-00013-f003:**
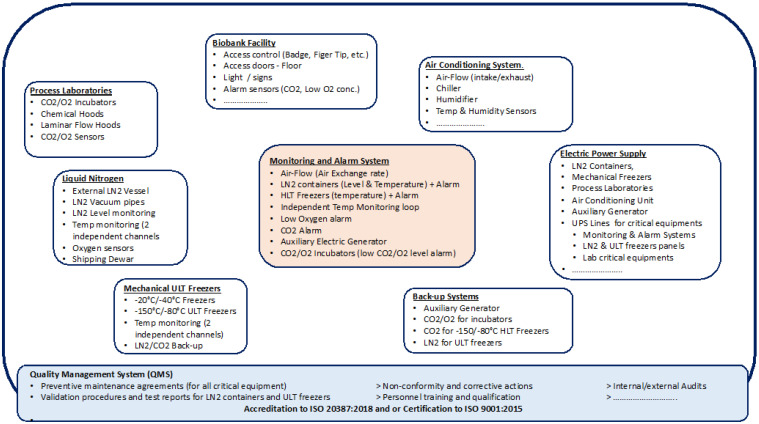
Biobanking infrastructure.

## Data Availability

Not applicable.

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
