# Peer review of "New Challenges for Biobanks: Accreditation to the New ISO 20387:2018 Standard Specific for Biobanks"

_biotech, 2021, doi:10.3390/biotech10030013_

Round 1

Reviewer 1 Report

The title of the manuscript is catchy but the text does not reflect it. What are the challenges? Why these challenges are new? What are the fundamental differences to already existing certification and accreditation standards?

The introduction as well as the main text need to be extended to include insights from Bioressource Centers (BRC) and culture collections. Those would also qualify as biobanks. But their functions are different from what is described in the manuscript.

Following the title, the major focus of the manuscript should be on new features of the ISO 20387.

The manuscript is presently too dry. It will benefit from examples explaining the ISO 20387 and other standards. Please note that readers of the manuscript might be interested in implementing ISO 20387. They need to know what to expect and understand whether they need the accreditation and at which costs.

Author Response

Please see the attachmnet

Reviewer 2 Report

It is a well-written article on a topic of interest to the entire biobank community. However, there are still a number of textual matters that need to be adjusted with regard to references to literature references and figures, typing errors, no or double spaces, uniform word use, etc. I hereby name a number of them:

  • The order of the references is not clear. It is neither in chronological order nor alphabetically.
  • References are included in the abstract. This is not common.
  • The word biobank is used in the text with and without a capital letter. Please use uniform spelling.
  • Abstract – background: “With the publication of the ISO 20387:2018 must of the biobanks…” must be “With the publication of the ISO 20387:2018 most of the biobanks…”
  • Introduction – paragraph 2: the term BMaD must be spelled out.
  • Introduction – paragraph 5: “Biobanks can vary not only…” must be “Biobank cannot only vary…”
  • The order of the Figures is incorrect. First Figure 2 is mentioned in the text. Figure 1 is not mentioned until later. After all, no reference is made to figure 3 in the text.
  • Discussion and Results – paragraph 1: “…are the first ISO standard focused on biobanking management. Implementation to the ISO 20387…” must be “…are the first ISO standards focused on biobanking management. Implementation of the ISO 20387…”
  • Discussion and Results – paragraph 1: QMS should be write out in full.
  • Conclusion – paragraph 1: “The objective for a biobank…” must be “The objectives for a biobank…”
  • References – number 8: “…Implementation Guidelines for ISO…” must be “…Implementation Guide for ISO…”

The author mentions in “Discussion and Results” that quality of samples can be the bottleneck for diagnosis and research studies in case of the BMaD are collected by several biobank which use different standards.” I miss mentioning the importance of standardizing the pre-analytical process and the lack of knowledge about this.

The author describes the scope of the ISO 20387:2018 (“Discussion and Results” and “Conclusions”). One of the objectives is to reduce costs. Is this about reducing research waste? Please elaborate.

In Figure 2 information regarding storage of samples can be found. However, information regarding storage at -20°C is missing. This also applies to the text below Figure 2 (“3. Biobank infrastructure”) and Figure 3.
